# Optimization of planting dates of Jalapeno pepper (*Capsicum annuum* 'Jalapeño' L.) and cantaloupe (*Cucumis melo* var. *cantalupensis* Ser.) relay cropped with strawberry (*Fragaria* × *ananassa* Duchesne)

**Ravneet K. Sandhu, Nathan S. Boyd**👤*, **Qi Qiu, Zhegfei Guan**

Gulf Coast Research and Education Center, University of Florida, Wimauma, Florida, United States of America

* nsboyd@ufl.edu

**Data Availability Statement:** All relevant data are within the manuscript and its Supporting Information files.

## Abstract

Florida strawberry growers frequently relay-crop (RC) vegetables with strawberry to offset high input costs. Relay-cropping consists of planting two crops at different times on the same raised bed which helps growers' lower risk due to diversification and maximize economic returns. Four separate experiments on relay cropping strawberry with jalapeño pepper and cantaloupe were conducted at Balm, Florida, in 2016–17 and 2017–18. The objective was to a) determine the effects of relay-cropping on crop yields, b) optimize pepper and cantaloupe planting date, and c) optimize the strawberry termination date when relay cropping with vegetables. Strawberry yields were unaffected by the presence of vegetables. Pepper yields were unaffected by the presence of strawberries, but cantaloupes yields were significantly (p = 0.0250) lower when planted with strawberry. Early January to early-February were the optimum dates to transplant pepper with strawberries transplanted in October but date of planting did not affect cantaloupe yield. Early January to early-February transplant dates for pepper resulted in net profits of around $23000-38000/ha in 2016–17 compared to the baseline treatment (strawberries alone). However, in 2017–18 all of the planting dates of pepper with strawberry except January 4 resulted in losses of $2000-18000/ha. In 2016–17, cantaloupes planted in late January and early February resulted in profits of $2986.3 and 2705.1/ha, respectively. All other planting dates resulted in loses compared to baseline treatment. In 2017–18, all the planting dates resulted in net profits of around $6700-14500/ha. In conclusion, cantaloupe and jalapeño pepper can be relay cropped with strawberry with no negative effects on strawberry yield. However, early planting dates tend to maximize economic return.

**Funding:** NSB received the grant number "USDA-AMS-SCBGP-2017" from Florida Department of Agriculture and Consumer Services. https://freshfromflorida.force.com/grants/s/ The funders had no role in study design, data collection and analysis, decision to publish, or preparation of the manuscript.

**Competing interests:** The authors have declared that no competing interests exist.

# Introduction

Strawberry (*Fragaria × ananassa* Duchesne) is a commonly consumed berry fruit crop worldwide. Globally, the United states rank second in strawberry production with California and Florida producing an estimated 91% and 8% of total US production, respectively [1]. With the introduction of NAFTA in 1994, the US produce industry faced challenges with increased competition from imported produce [2]. The lower production costs and the availability of cheap labor outside the US have affected the profitability of the strawberry industry across the US [3]. In addition, strawberry is grown only as a winter crop in Florida unlike California which has a much longer production cycle that can overlap with Florida production and reduce berry prices [4]. Competition from foreign and domestic markets coupled with high production costs, volatile prices, and low labor availability continues to have a negative impact on the Florida strawberry industry [3, 5, 6].

One of the potential solutions to address these challenges is to grow more than a single crop on strawberry beds. Double cropping of melons with strawberries at the end of the season is already well established in Florida. However, many other vegetable crops can also be planted as a relay crop with strawberries and provide income stability, crop diversification, and can also provide a wider range of planting dates for growers that rely on contractual sales. Relay cropping is the practice of planting the second crop at different times on the raised beds where the main crop is already growing [7]. As the two crops are temporally segregated, so, the resource use efficiency is less likely to be affected by competition between two crops [8]. Moreover, relay cropping can enhance resource utilization in terms of using the same field preparation, fumigation, plastic, drip tape, and fertilizer for both crops [7]. Double cropping had shown to recover the 60% of total strawberry production costs of an annual plasticulture system when strawberries were double cropped with melons [9]. However, relay cropping of melons (secondary crop) with strawberries (main crop) in March resulted in reduced yields due to high temperature conditions during the flowering stage [10]. For really croppig to be successful it is critical to determine the optimum planting date for secondary crops when relay cropping with strawberries because competitive interactions and extreme weather (temperature) could limit secondary crop yields.

Jalapeno pepper (*Capsicum annuum* 'Jalapeño' L.) is a warm season crop from the Chile pepper group which also includes crops such as paprika, Anaheim and banana peppers. The United States ranks 5th in the production of Chile peppers with a production value of $135 million dollars in 2015 with an area harvested of 18,100 acres [11]. California, New Mexico, and Texas are the leading hot pepper producing states. However, Florida growers also grow hot peppers as a niche crop [12]. The optimum temperature requirement for peppers is 21-26º C. Peppers are sensitive to extreme low and high temperatures and temperatures below 12 ºC or above 33 ºC can result in poor fruit set [13]. Knowing that the weather plays major role in determining the planting date, the early planting dates could add growth competition from main crop. However, too late planting of peppers will affect yields due to low temperature in winter months.

Cantaloupe (*Cucumis melo* var. cantalupensis Ser.) is another popular warm-season crop grown from January-May in Florida. The estimated total production value for cantaloupes is $10.3 million with 2200 acres of harvested area [11]. Similar challenges exist with cantaloupes relay cropped with strawberries. The weather indirectly affects the yields of cantaloupes by altering bee activity [14]. Therefore, it is very important to optimize the planting date for both crops when growing as a secondary crop with strawberries. Planting date also determines the length of time both crops occur on the bed at the same time and as a result the duration of inter-specific competition.

**Table 1. Average monthly air temperature (˚C) across 2010–2018 at GCREC, Balm.**

| Year | Temperature (˚C) | | | | | | | | | | | |
|---|---|---|---|---|---|---|---|---|---|---|---|---|
| | January | February | March | April | May | June | July | August | September | October | November | December |
| 2010 | 11.4 | 11.6 | 14.8 | 20.6 | 25.1 | 26.9 | 26.9 | 26.8 | 26 | 22.2 | 18.7 | 10 |
| 2011 | 13.8 | 16.8 | 18.9 | 22.7 | 24.2 | 26.1 | 26.6 | 27.2 | 25.9 | 21.5 | 19.4 | 17.9 |
| 2012 | 15.6 | 19 | 21.5 | 21.8 | 25.4 | 25.6 | 26.7 | 26.5 | 25.6 | 22.8 | 17.1 | 17.4 |
| 2013 | 18.2 | 17.4 | 15.3 | 22.6 | 23.2 | 25.9 | 25.8 | 26.6 | 25.5 | 23.3 | 20.5 | 19.4 |
| 2014 | 14 | 18.1 | 18.4 | 21.5 | 24.3 | 26.3 | 26.9 | 27.3 | 25.5 | 22.8 | 17 | 17.1 |
| 2015 | 15.9 | 14.7 | 21.5 | 24.1 | 25.3 | 26.5 | 26.6 | 26.9 | 26.5 | 23.8 | 22.8 | 21.7 |
| 2016 | 14.9 | 15.9 | 21 | 22.3 | 24.4 | 26.9 | 27.6 | 26.9 | 26.4 | 23.3 | 19.3 | 19.6 |
| 2017 | 17.3 | 19 | 19.1 | 23.1 | 24.8 | 25.7 | 26.7 | 27 | 26.1 | 23.4 | 20.1 | 17.5 |
| 2018 | 13.5 | 21.2 | 17.3 | 21.8 | 23.9 | 26.7 | 26.9 | 26.7 | 27.1 | 24.9 | 20.5 | 17.7 |

We hypothesize that the two crops growing together will compete for resources and as a result earlier planting dates for the secondary crop should result in lower yields due to longer periods of inter-specific competition. The objectives of the study are to, a) determine if competition between the primary and secondary crop occurs in terms of yields and morphological characteristics, b) determine the optimal planting date for the secondary crop that optimizes yield, i.e. peppers and cantaloupes, c) determine the effect of strawberry termination date on the yield of the secondary crop, and d) determine the effect of planting date on the economic return when relay cropping cantaloupes and peppers with strawberries.

## Materials and methods

### Experimental setup

All field experiments were performed at the Gulf Coast Research and Education Center (27˚N, 82˚W), University of Florida, in Balm, FL. The type of soil at the location was a Myakka fine sand (sandy, siliceous, hyperthermic Oxyaquic Alorthod). The soil texture was 98% sand, 1% silt, and 1% clay, and composition was 1.5% organic matter with pH of 5.5–6. The temperature and rainfall data at the research site is presented in Tables 1 and 2. Raised beds were formed with a bed top width of 66 cm and height of 30.5 cm on September 10 and August 15, in 2016 and 2017, respectively, with pressing equipment (Kennco Manufacturing, Ruskin, FL). The beds were fumigated with 63.4% 1, 3-dichloropropene and 34.7% chloropicrin (Telone® C35; Dow AgroSciences, Indianapolis, IN) at 341 kg/ha. Drip tape was placed in the middle of the beds (Jain Irrigation Inc., Haines City, FL) with a flow rate of 0.95 L/min and emitters every 30

**Table 2. Monthly total rainfall in cm across 2010–2018 at GCREC, Balm.**

| Year | Rainfall (cm) | | | | | | | | | | | |
|---|---|---|---|---|---|---|---|---|---|---|---|---|
| | January | February | March | April | May | June | July | August | September | October | November | December |
| 2010 | 8.1 | 5.6 | 15.6 | 7.1 | 2.3 | 21.0 | 18.5 | 34.3 | 8.7 | 0.0 | 3.1 | 1.3 |
| 2011 | 10.5 | 1.2 | 17.5 | 2.4 | 2.7 | 12.3 | 23.1 | 22.3 | 6.4 | 10.9 | 1.5 | 0.8 |
| 2012 | 2.3 | 1.4 | 2.1 | 3.5 | 4.7 | 37.8 | 6.6 | 31.3 | 10.4 | 8.4 | 0.3 | 6.2 |
| 2013 | 0.8 | 2.6 | 2.6 | 10.6 | 9.1 | 42.0 | 22.1 | 17.7 | 13.9 | 4.5 | 1.4 | 3.8 |
| 2014 | 7.3 | 0.7 | 13.2 | 3.4 | 14.9 | 10.9 | 26.3 | 8.6 | 30.4 | 5.5 | 18.3 | 2.4 |
| 2015 | 1.7 | 13.9 | 3.5 | 8.1 | 0.6 | 25.8 | 40.3 | 26.6 | 9.2 | 3.4 | 7.1 | 1.9 |
| 2016 | 17.0 | 4.2 | 4.9 | 5.7 | 25.5 | 24.0 | 17.6 | 27.2 | 10.1 | 2.6 | 0.1 | 0.5 |
| 2017 | 3.2 | 4.0 | 1.8 | 7.0 | 4.5 | 26.2 | 24.7 | 19.9 | 22.7 | 1.5 | 0.3 | 5.3 |
| 2018 | 8.7 | 1.8 | 2.9 | 8.5 | 32.9 | 17.5 | 27.7 | 20.1 | 11.6 | 1.4 | 5.5 | 21.2 |

cm. Black plastic mulch, virtually impermeable film (VIF) in 2016 and totally impermeable film (TIF) in 2017 (Berry Plastics Corp., Evansville, IN) was used to cover the raised beds. Industrial recommendations were followed for fertilization, disease and insect pest management [15, 16].

Four separate experiments a) strawberry relay cropped with peppers, b) strawberry relay cropped with cantaloupes, c) strawberry termination date when relay cropped with peppers, and d) strawberry termination date when relay cropped with cantaloupes were setup in 2016 and 2017. Strawberry (cv. Radiance$^{TM}$) was transplanted on 6m long plots in two parallel rows with 38 and 30 cm spacing between plants and rows on October 12 and 10, in 2016 and 2017, respectively. Strawberry plants were irrigated with an overhead irrigation system for 10 to 15 days after transplanting to reduce heat stress. Fertilization of strawberries was managed according to the industrial recommendations. Fertilizer (6-2-4 N-P$_2$O$_5$-K$_2$O) was applied (fertigation) through the drip tape three times weekly at a rate of 150 kg N per ha [15].

Cantaloupes (*Cucumis melo* var. *cantalupensis* Ser.) and Jalapeno peppers (*Capsicum annuum* 'Jalapeño' L.) were planted in the center of bed and spacing between plants was kept standard at 61 cm and 38 cm, respectively as per UF/IFAS recommendations [17, 12]. Once, the strawberry crop was terminated, the standard recommended fertilization prcatices were followed for peppers and cantaloupes. The fertilizers were applied at the rate of 226-112-112 (N-P-K) kg/ha and 170-112-112 (N-P-K) kg/ha through drip irrigation for peppers and cantaloupes, respectively [15]. All the crops were irrigated for 30–40 minutes twice a day.

At the end of the season, strawberry plants were terminated via hand-pulling both years. Strawberries were terminated in the first week of March for trial a and b.

**Strawberries relay cropped with peppers or cantaloupes.** The experiments were set up as a 2x6 factor factorial design in a randomized complete block with four replications. The experiment was conducted in two growing (fall/spring) seasons' 2016–17 and 2017–18. The first factor was the transplanting date of the secondary crop: January 4, January 18, February 1, February 15, and March 1, with a non-planted control in spring 2017 and 2018. The second factor was the presence or absence of strawberry plants.

**Strawberry termination.** Two separate experiments; one with canteloupes and one with peppers as the relay crop, were set up same as previous experiment (CRBD) with four replications in two growing seasons (2016–17 and 2017–18). Cantaloupes or peppers were planted on February 1 (as per grower practices) as the relay crop in all the treatments in both years. The main treatment effect was the strawberry termination date, conducted at weekly intervals (February 7, February 14, February 21, February 28 and March 7, and a non-planted control). The non-planted control was only peppers or cantaloupes with no strawberry plants.

## Data collection

Strawberry yield data were collected from the first week of January until the first week of March. Berries were hand-picked, weighed and counted twice a week. Photosynthesis was measured once in early March which corresponds to the end of the strawberry season with a LICOR-6400XT portable photosynthesis system (LICOR Biosciences, Lincoln, NE). Brix measurements were recorded on strawberries growing alone and with peppers and cantaloupes by using a pocket refractometer (PAL-1, ATAGO Co., LTD. Tokyo, Japan). Pepper height and cantaloupe vine length were recorded every fifteen days after transplanting up to 60 days. The pepper and cantaloupe fruit were hand-picked and weighed once a week from May to the first week of June. Same number of harvestings were done for secondary crops planted at different dates. Pepper plants and cantaloupe vines were terminated by applying paraquat (Gramaxone SL 2.0; Greensboro, NC) to all of the trials at the end of the season.

## Economic analysis

Strawberry, pepper and cantaloupe price per kilograms were estimated by using everyday prices collected from the United States Department of Agriculture (USDA) Agricultural Marketing Service. Due to the market price volatility of all the crops, the average price for five successive years was used. The average revenue for each treatment was determined by the multiplication of price and yield per harvest.

Input unit cost was based on Florida strawberry production costs and trends [5]. Supplementary cost assessments were also done [18] and the input prices were updated by using the Producers Price Index (PPI) [19]. Pepper and cantaloupe costs were recorded based on a survey among growers by Dr. Guan and Dr. Wu. The prices of Glyphosate and Gramaxone in the respective years of the experiment were collected from local suppliers. The standard farm labor income rate was acquired for farmhands, laborers, nursery, crops, and greenhouse [19].

## Data analysis

The data was analyzed using the PROC Glimmix procedure in SAS (Version 9.2; SAS Institute, Inc., Cary, NC) for the yield data. Assumptions of ANOVA were checked. No data transformations were done. Replication and year were specified as a random effect, while the year, date of planting and presence or absence of strawberries was considered as fixed factors. Tukey's honest significant difference test ($\alpha = 0.05$) was used to separate the means. In the strawberry termination experiments, the block was considered a random factor and the date of strawberry termination was considered as a fixed factor. Pepper height and cantaloupe vine length data were subjected to linear regression in sigma plot$^{TM}$ (Systat Software Inc., San Jose, CA). Student's independent t-tests were used to compare the slopes of the height of the secondary crop [20].

## Results and discussion

### Strawberry yields

Strawberry yields in 2017–18 were significantly ($p < 0.0001$) higher than 2016–17 with average values of 33681 and 25800 kg/ha, respectively when relay cropped with peppers. Strawberry yields were not affected by the presence of peppers in 2016–17 ($p = 0.3540$) and 2017–18 ($p = 0.3506$). Similarly, in the cantaloupe study, strawberries yield in 2017–18 was significantly ($p < 0.0001$) higher than strawberries produced in 2016–17 with average values of 27761 and 23757 kg/ha, respectively. Similarly, strawberry yields were unaffected by the presence of cantaloupes in 2016–17 ($p = 0.0641$) and 2017–18 ($p = 0.0712$). Strawberries were planted in October and grew by themselves for 3 months which is critical growth period for strawberries [21] and consequently, there was no inter-specific competition during the critical growth period of strawberries. The lack of yield affect can likely be attributed to the early establishment of strawberry which would provide a competitive advantage. Another important factor could be the shade tolerant behavior of strawberry plants. Previous research has shown that although shade affects the growth and development of strawberry plants, there was no effect on the strawberry yields [22]. Similar relay cropping research conducted in Florida reported relay cropping with eggplants, muskmelons, cucumbers, and squash had no effects on strawberry yields [10, 23, 24].

There was no difference in brix % of strawberries planted with cantaloupes ($p = 0.7019$) and Jalapeno peppers ($p = 0.4870$) compared to strawberries planted alone. The photosynthesis rate was similar in case of strawberry plants grown with and without Jalapeno peppers ($p = 0.5576$) and Cantaloupes ($p = 0.0630$). A study on strawberry interarcropped with

different vegetables reported the results on strawberry brix %, which are in agreement to the results in current study [25]. The sugar accumulation in the fruits is mainly affected by photosynthesis [26]. So, given the fact that there was no effect in photosynthesis rate of strawberry plants, the sugar content and fruit color did not vary among the treatments.

## Peppers relay-cropped with strawberries

Pepper and cantaloupe yields differed between seasons (p<0.0001). Consequently, yield data is presented separately for each year. Crop yields were significantly higher in 2017–18 than 2016–17. In 2017–18 TIF plastic mulch was used which is known to increase the soil temperature at 30 cm depth [27] which may partially explain the difference. Kumar et al. (2002) reported that mulches increase the soil temperature and benefit some crops in terms of yield, root growth and nutrient uptake [28].

**Pepper heights.** Pepper height and year interaction were not significant and consequently the two-year data were pooled and analyzed together. The presence of strawberries significantly reduced the rate of change in height over time (growth rate) of peppers (p = 0.0383) (Fig 1). Peppers growing with strawberries could have restricted root growth due to allelopathic effect of strawberry roots [29], which is known to reduce leaf and shoot growth in solanaceae crops [30, 31]. However, the pepper transplant date did not have an effect on the pepper growth rate (p = 0.2403). These results signify that the presence of the strawberry crop hindered pepper growth and yields but transplant date did not play a significant role in terms of pepper growth.

**Pepper yields.** Pepper yields in 2016–17 were significantly (p<0.0001) higher than peppers produced in 2017–18 with average values of 35791 and 18224 kg/ha, respectively, due to plant mortality in harsh (freeze) weather conditions in the second season. In both the years, interaction of planting date and presence/absence of strawberries did not affect fruit number and weight (p = 0.7723 and p = 0.8577 in 2016 and 2017, respectively). Presence of strawberries did not affect the pepper yields (p = 0.3961 and p = 0.2978 in 2016 and 2017, respectively). Pepper yields were 33491 and 38092 kg/ha when planted with and without strawberries averaged over planting dates, respectively. The date of planting had significant (p = 0.0026 and p = 0.0005) in 2016 and 2017, respectively) effect on pepper yields in both the years (Fig 2A). Yields were significantly higher at earlier planting dates than later planting dates. Overall pepper yields were significantly higher when planted on January 4, January 18, and February 1 than the peppers planted on February 15 and March 1 (Fig 2B).

This would suggest that weather conditions are a more important variable than competition duration and later in the season when peppers started producing fruits the strawberry plants had already been terminated, so early season competition had no effect on yield. The flowering months in peppers planted in late February and early March coincided with high daily temperatures (>29.0ºC), which could hinder fruit yields in Solanaceae crops [32]. It has been reported that average high temperatures of >32 Cº and average low temperatures of <21 Cº resulted in low fruit setting and high temperatures also have a direct effect on the pollen health, hormonal imbalances, and result in loss of carbohydrates [33, 34]. Hence, the overall yield of peppers transplanted on later planting dates was reduced.

## Cantaloupes relay-cropped with strawberries

**Cantaloupe vine length.** Vine length data was pooled across years due to insignificant seasonal interactions. Cantaloupe vine growth rate in terms of the slope was not affected by the presence orabsence of strawberries (p = 0.4402) and by date of the planting of cantaloupes (p = 0.3887) (Fig 3). These results suggest that cantaloupes are less susceptible to strawberry competition and temperature extremes than peppers.

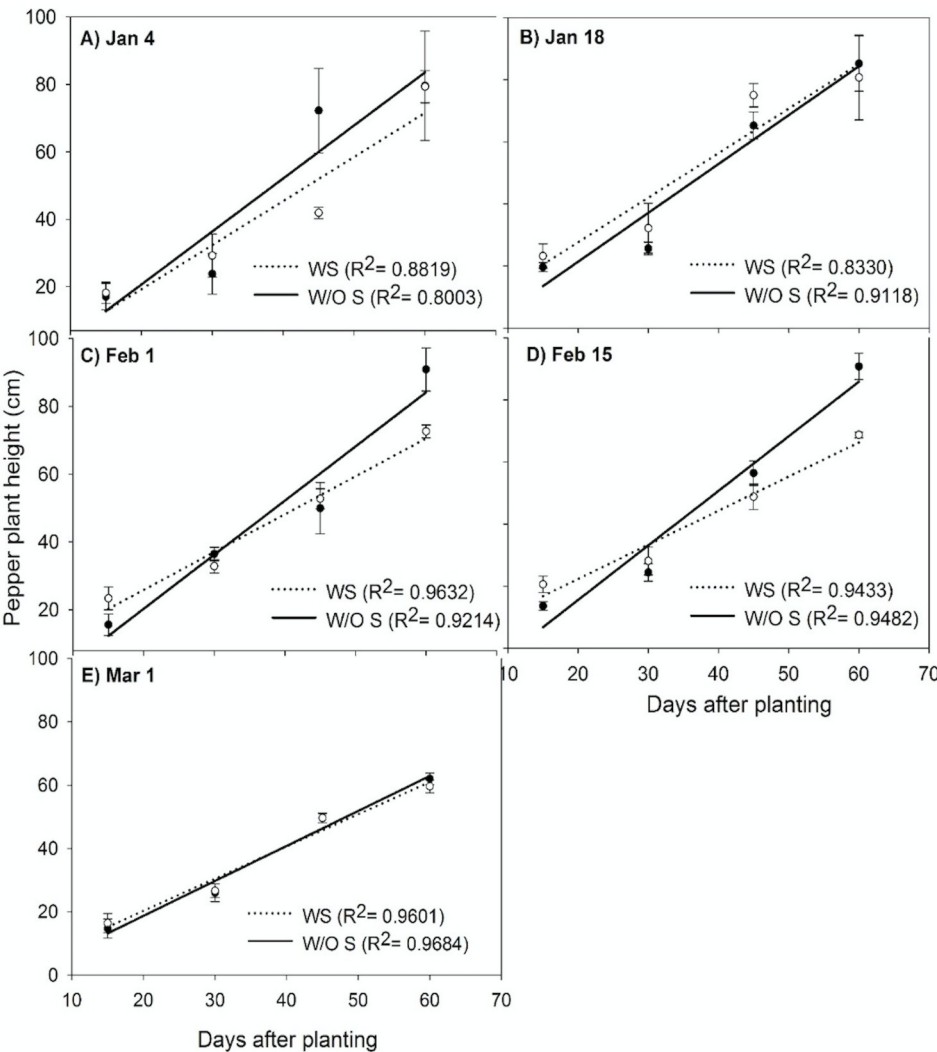

**Fig 1.** Pepper heights A) Jan 4, B) Jan 18, B) Feb 1, C) Feb 15 and D) Mar 1 at GCREC in 2016–17 and 2017–18. Data are mean ± SE (WS- With strawberries and W/O S–Without strawberries) with two years pooled together recorded at 15, 30, 45, 60, and 75 DAT (Days after transplanting).

**Cantaloupe yield.** Cantaloupe yields were significantly ($p < .0001$) higher in 2017–18 than 2016–17 with an average yield of 58769 and 32316 kg/ha, respectively. The difference between years could be largely attributed to a greater number of harvests in 2017–18 then the previous season. In 2017–18, the date of planting of cantaloupes did not affect ($p = 0.0905$) cantaloupe yields. However, the yields of cantaloupes were significantly ($p = 0.0250$) higher when planted without strawberries (Fig 4).

The fruit (size) of cantaloupe crop is dominant sink compared to the whole plant and cucurbits exhibit high growth rates in general [35, 36], and require a vast amount of nutrients and water during early growth periods. It could be argued that strawberries competed with cantaloupes for water and nutrients and low nutrient status and water stress are known to decrease cantaloupe yields significantly [37]. Although the current study did not evaluate the nutritional status of cantaloupes relay cropped with strawberries. However, some of the previous studies reported that muskmelons intercropped with fennel did not have difference in fruit quality indices like fructose, sucrose and glucose content compared to monocropped

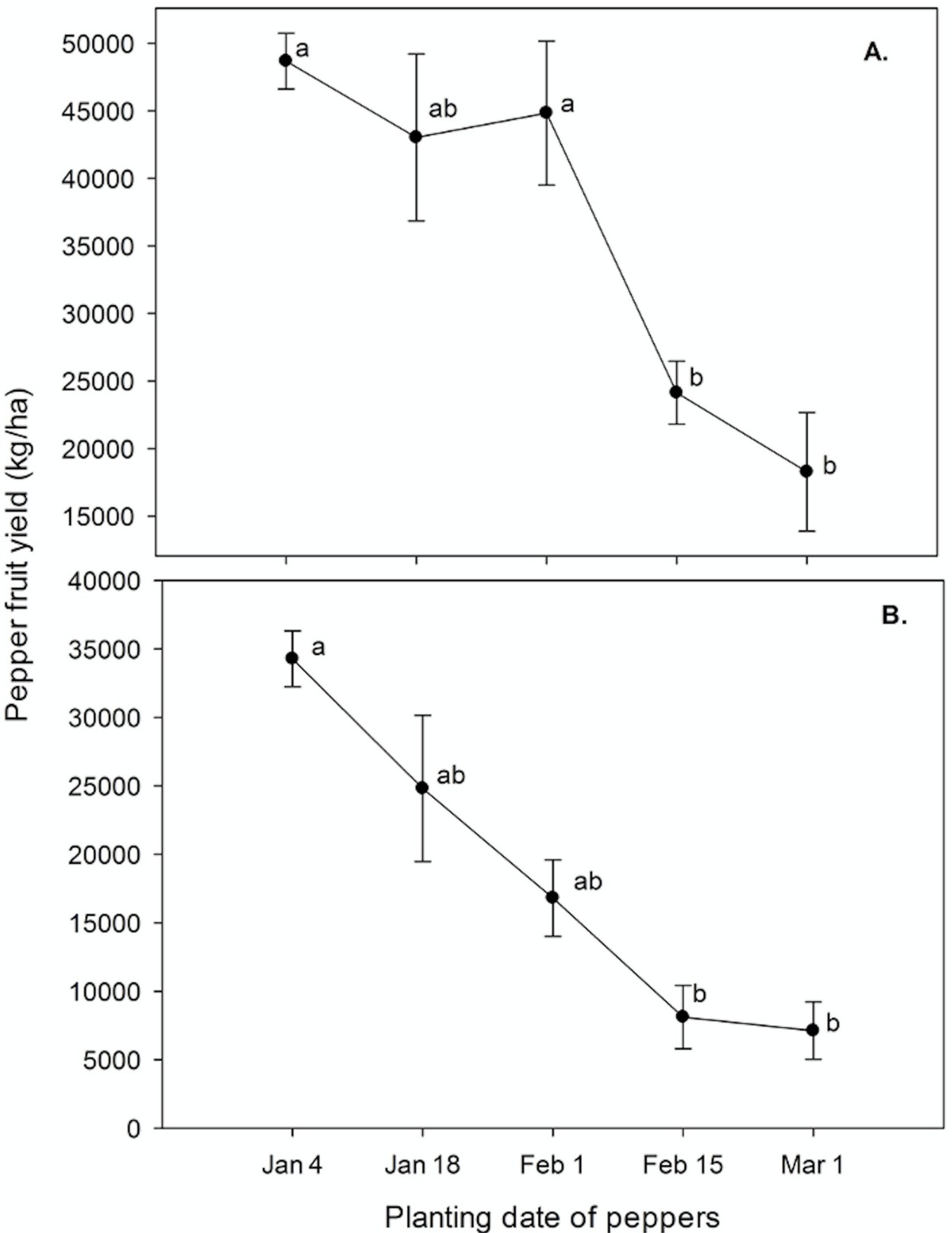

**Fig 2.** Pepper yield averaged across transplant dates at GCREC in A. 2016–17, B. 2017–18. Data are mean ± SE of strawberry yield of two years pooled together. Letters above each point denote significant differences (p<0.05).

muskmelons [38]. They further reported that muskmelons intercropped with other crops like tillered-onions and wormwood did have reduced amounts of glucose, fructose and sucrose levels compared to muskmelons planted alone. This could be the effect of interspecific competition, because muskmelons do not have stored starch like other fruits and require continuous supply of photoassimilated from leaves to produce sugars [26]. So, competition with crop could reduce the photoassimilate accumulation in the leaves and reduce the sugar content in fruits.

In 2016–17, the planting date (p = 0.6848) of cantaloupes and the presence of strawberries (p = 0.4443) affected the cantaloupe fruit weight. A study reported on muskmelon relay cropped with strawberries subjected to different summer fallows reported contrasting results with no effects on muskmelon yields [23]. However, contrasting results were reported in a study where muskmelon planted in march had reduced yields due to high temperatures [10]

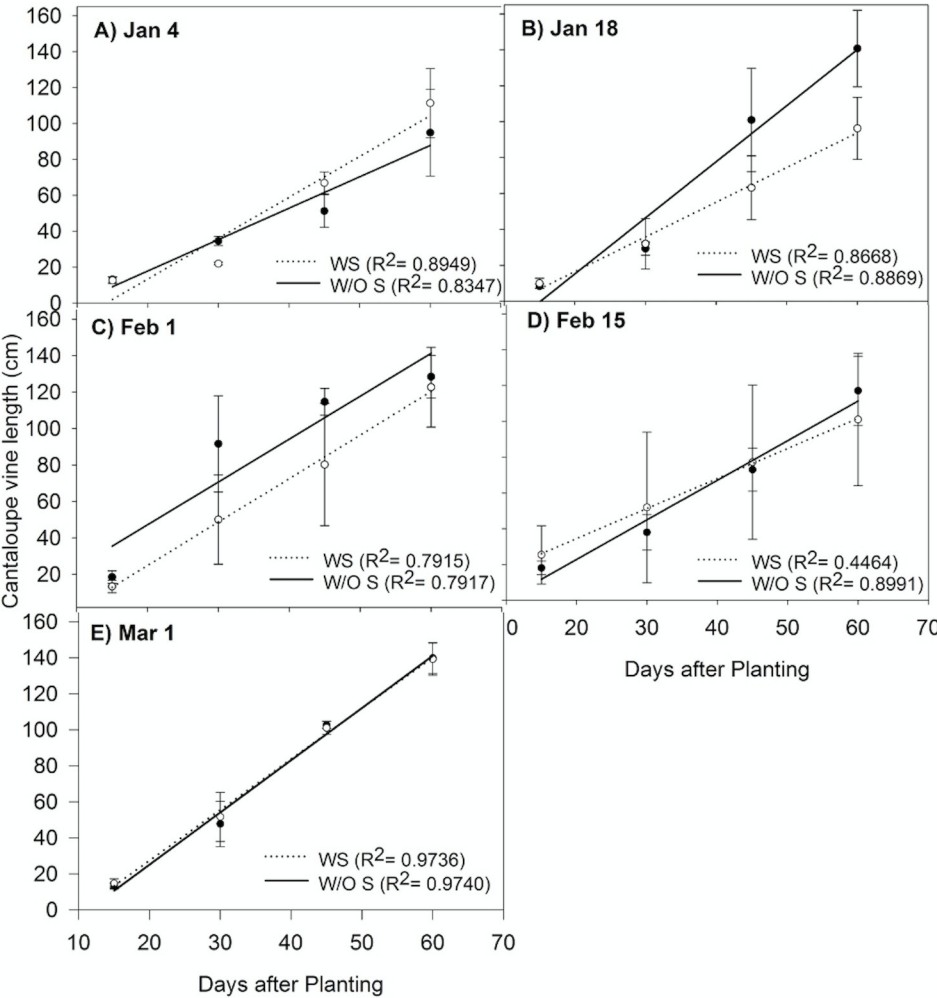

**Fig 3.** Cantaloupe vine length A) Jan 4, B) Jan 18, B) Feb 1, C) Feb 15 and D) Mar 1 at GCREC in 2017 and 2018. Data are mean ± SE (WS- With strawberries and W/O S–Without strawberries) with two years pooled together recorded at 15, 30, 45, 60, and 75 DAT (Days after transplanting).

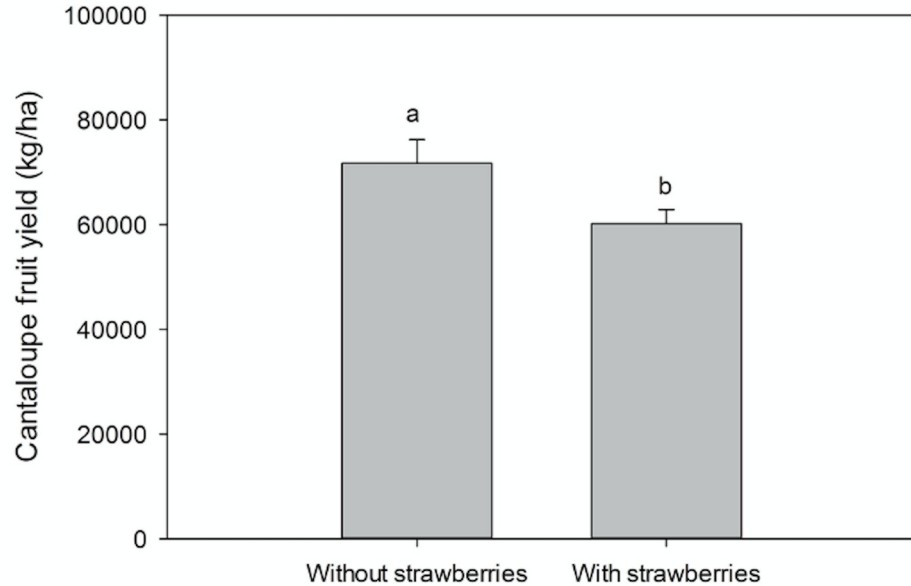

**Fig 4. Overall cantaloupe yield planted with and without strawberries at GCREC in 2017–18.** Data are mean ± SE of strawberry yield of two years pooled together. Letters above each point denote significant differences (p<0.05).

but the temperature never reached high enough to reduce yields in the present study. This indicates that competition is more important in the case of cantaloupe reproductive growth as compared to vegetative growth and vice-versa in the case of peppers.

## Strawberry termination

Strawberry termination did not affect the yields, heights or vine length of peppers and cantaloupes, which means early release from competition had no beneficial effects. These results support our theory that competition duration does not affect the secondary crop yields as proven in other experiments and extending the strawberry season is not detrimental for the secondary crop. Our results determine that the timing of crop termination need to be based on yields, berry quality, and markets not based on enhancing yields of the second crop. Moreover, early termination will leave the planting holes open which can promote weed seed germination and growth and hinder end of season plastic removal. Early strawberry termination could be an option for growers if there is no strawberry market available. However, terminating strawberries to avoid secondary crop yield losses would lead to revenue losses from strawberries.

## Net economic revenue analysis

**Peppers relay cropped with strawberries.** The additive costs of fumigation, plastic mulch, and drip system were $3509.9 per hectare for strawberry production and this cost will be shared between two crops when relay cropping with peppers. The high additional costs are pepper transplant and picking cost at $4838 and $3837.5, respectively when peppers were relay cropped with strawberries (Table 3).

In 2016–17, strawberry relay-cropping with peppers resulted in net profits compared to baseline treatment on all planting dates except for March 1. Net profit was highest when peppers were planted with strawberries on January 4 ($38948.5/ha) followed by February 1 ($22352.3/ha), January 18 ($24827.2/ha) and February 15 ($9592.1/ha) compared to baseline

**Table 3. The major cost items of the baseline treatments and pepper treatments.** The dashed line shown in the table means there is no difference.

| Input Names | Strawberry production costs ($/ha) | Change in costs associated with relay cropped pepper ($/ha) | Differences between baseline and pepper only treatments ($/ha) |
|---|---|---|---|
| Fumigant (C35/Kpam) | 1931.14 | 0 | 0 |
| Fungicide | 1594.07 | 0 | -1594.07 |
| Herbicide | 351.32 | 0 | -351.32 |
| Insecticide | 1377.96 | 0 | -1377.96 |
| Seed and Transplants | 6811.17 | 0 | -6811.17 |
| Tractor: Fuel cost | 1638.86 | 0 | -1638.86 |
| Tractor: Lubrication cost | 163.89 | 0 | -163.89 |
| Drip Tape | 571.51 | 0 | 0 |
| Fertilizer | 3585.66 | 0 | -3585.66 |
| Glyphosate | 27.51 | 0 | -27.51 |
| Gramaxone | 37.24 | -8.4 | -37.24 |
| Plastic Mulch | 1007.32 | 0 | 0 |
| Waste Disposal | 287.06 | 0 | -287.06 |
| Bedding | 508.30 | 0 | -508.30 |
| Planting | 1008.26 | 0 | -1008.26 |
| Cutting, Hoeing & Hand weeding | 2706.45 | 0 | -2706.45 |
| Spraying | 117.60 | 0 | -117.60 |
| Picking (at peak time) | 24586.60 | 0 | -24586.60 |
| End of season clean up labor | 978.46 | 0 | -978.46 |
| Cooling | 3674.81 | 0 | -3674.81 |
| Packing materials | 10622.49 | 0 | -10622.49 |
| Disking | 72.04 | 0 | – |
| Peppers transplants | 0 | 4838.32 | 4838.32 |
| Strawberry hand pull cost | 0 | 296.53 | 0 |
| Herbicides (Pepper) | 0 | 118.61 | 118.61 |
| Fungicide & insecticide (Pepper) | 0 | 2221.48 | 2221.48 |
| Fertilizer (Pepper) | 0 | 1937.30 | 1937.30 |
| Stake & tie (Pepper) | 0 | 3961.10 | 3961.10 |
| Tractor: Fuel cost (Pepper) | 0 | 159.63 | 159.63 |
| Picking Cost (Pepper) | 0 | 3837.55 | 3837.55 |
| Packing costs (Pepper) | 0 | 2683.56 | 2683.56 |
| Clean-up (Pepper) | 0 | 1821.17 | 1821.17 |
| Cooling costs (Pepper) | 0 | 741.32 | 741.32 |
| Total | 63659.72 | 22608.17 | -24855.22 |

treatment (only strawberries). The lower yields due to high temperature conditions in May/June (corresponds to flowering of peppers) resulted in the lowest yields and hence peppers planted with or without strawberries on March 1 resulted in net losses compared to the baseline treatment. Peppers planted alone on January 4, 18, and February 15 resulted in a net revenue of $15236.2, $18867.7, and $19324.4 per ha, respectively (Table 4), which is lower than the net returns gained by relay-cropping of strawberries with peppers on early planting dates. The total overall revenue from strawberry relay cropped with pepper higher than pepper alone which shows that strawberry revenue is very important to gain the highest overall income from the relay cropping system.

**Table 4. The total cost and revenue comparison between strawberry productions versus strawberry relay cropped with peppers in 2016–17.** A negative sign indicates reduced costs, returns and profits compared to strawberries alone.

| Treatments* | Added costs of the alternative treatment ($/ha) | Reduced returns of the alternative treatment ($/ha) | Total negative effects of the alternative treatment ($/ha) | Reduced costs of the alternative treatment ($/ha) | Added returns of the alternative treatment ($/ha) | Total positive effects of the alternative treatment ($/ha) | Net effects relative to treatment 1 ($/ha) |
|---|---|---|---|---|---|---|---|
| Only strawberries | 0.0 | 0.0 | 0.0 | 0.0 | 0.0 | 0.0 | 0.0 |
| S+P (Jan 4) | 21639.5 | 0.0 | 21639.5 | 0.0 | 60588.1 | 60588.1 | 38948.5 |
| S+ P (Jan18) | 21639.5 | 0.0 | 21639.5 | 0.0 | 43991.8 | 43991.8 | 22352.3 |
| S+ P (Feb 1) | 21639.5 | 0.0 | 21639.5 | 0.0 | 46466.7 | 46466.7 | 24827.2 |
| S+ P (Feb 15) | 21639.5 | 0.0 | 21639.5 | 0.0 | 31231.7 | 31231.7 | 9592.1 |
| S+ P (Mar 1) | 21639.5 | 0.0 | 21639.5 | 0.0 | 16113.7 | 16113.7 | -5525.8 |
| P (Jan 4) | 0.0 | -78898.2 | 78898.2 | -38726.3 | 55408.1 | 94134.5 | 15236.2 |
| P (Jan18) | 0.0 | -78898.2 | 78898.2 | -38726.3 | 59039.6 | 97765.9 | 18867.7 |
| P (Feb 1) | 0.0 | -78898.2 | 78898.2 | -38726.3 | 59496.3 | 98222.6 | 19324.4 |
| P (Feb 15) | 0.0 | -78898.2 | 78898.2 | -38726.3 | 25633.8 | 64360.1 | -14538.2 |
| P (Mar 1) | 0.0 | -78898.2 | 78898.2 | -38726.3 | 26871.6 | 65597.9 | -13300.3 |

*S- Strawberry and P- Peppers

[a]Net effects are the profits compared to baseline (only strawberries) treatment

In 2017–18, only peppers planted on January 4 with strawberries resulted in a net profit of $11449.7/ha compared to the baseline treatment. All other treatments (peppers with and without strawberries) resulted in losses ranging from $2300.0 –$47500.0/ha (Table 5) compared to baseline treatment. The two years' profits are based upon the yield results, which means that relay-cropping of peppers with strawberries is not always profitable. Relay cropping is profitable when peppers are planted in early to mid January with strawberries.

**Table 5. The total cost and revenue comparison between strawberry productions versus strawberry relay cropped with peppers in 2017–18.** A negative sign indicates reduced costs, returns and profits compared to strawberries alone.

| Treatments* | Added costs of the alternative treatment ($/ha) | Reduced returns of the alternative treatment ($/ha) | Total negative effects of the alternative treatment ($/ha) | Reduced costs of the alternative treatment ($/ha) | Added returns of the alternative treatment ($/ha) | Total positive effects of the alternative treatment ($/ha) | Net effects relative to treatment 1 ($/ha) |
|---|---|---|---|---|---|---|---|
| Only strawberries | 0.0 | 0.0 | 0.0 | 0.0 | 0.0 | 0.0 | 0.0 |
| S+P (Jan 4) | 21639.5 | 0.0 | 21639.5 | 0.0 | 33089.2 | 33089.2 | 11449.7 |
| S+ P (Jan18) | 21639.5 | 0.0 | 21639.5 | 0.0 | 20288.3 | 20288.3 | -1351.2 |
| S+ P (Feb 1) | 21639.5 | 0.0 | 21639.5 | 0.0 | 13141.5 | 13141.5 | -8498.0 |
| S+ P (Feb 15) | 21639.5 | 0.0 | 21639.5 | 0.0 | 5442.2 | 5442.2 | -16197.3 |
| S+ P (Mar 1) | 21639.5 | 0.0 | 21639.5 | 0.0 | 4854.6 | 4854.6 | -16784.9 |
| P (Jan 4) | 0.0 | -94551.7 | 94551.7 | -38726.3 | 33812.8 | 72539.2 | -22012.5 |
| P (Jan18) | 0.0 | -94551.7 | 94551.7 | -38726.3 | 33171.3 | 71897.6 | -22654.1 |
| P (Feb 1) | 0.0 | -94551.7 | 94551.7 | -38726.3 | 19330.4 | 58056.7 | -36495.0 |
| P (Feb 15) | 0.0 | -94551.7 | 94551.7 | -38726.3 | 9970.6 | 48696.9 | -45854.8 |
| P (Mar 1) | 0.0 | -94551.7 | 94551.7 | -38726.3 | 8386.1 | 47112.4 | -47439.3 |

*S- Strawberry and P- Peppers

[a]Net effects are the profits compared to baseline (only strawberries) treatment.

**Cantaloupes relay cropped with strawberries.** The fertilizer cost of strawberries was 83.4% higher than cantaloupes. The major additional input costs included in of relay cropping cantaloupe with strawberries production were Fungicide, picking cost and packing cost of $2668.7, $6699.3, $4828.9 per ha, respectively (Table 6).

The cantaloupe yield differrence between two years was clearly reflected in the economic analaysis results. In 2016–17, cantaloupes relay-cropped with strawberries on February 1 and 15 resulted in net profits of $3282.8 and $3001.6 per ha, respectively. All other planting dates

**Table 6. The major cost items of the baseline treatments and cantaloupes treatments.** The dashed line shown in the table means there is no difference.

| Input Names | Strawberry production costs ($/ha) | Change in costs associated with relay cropped cantaloupe ($/ha) | Differences between baseline and cantaloupe only treatments ($/ha) |
|---|---|---|---|
| Fumigant (C35/Kpam) | 1931.14 | 0 | 0 |
| Drip Tape | 571.51 | 0 | 0 |
| Fungicide | 1594.07 | 0 | -1594.07 |
| Herbicide | 351.32 | 0 | -351.32 |
| Insecticide | 1377.96 | 0 | -1377.96 |
| Seed and Transplants | 6811.17 | 0 | -6811.17 |
| Tractor: Fuel cost | 1638.86 | 0 | -1638.86 |
| Tractor: Lubrication cost | 163.89 | 0 | -163.89 |
| Fertilizer | 3585.66 | 0 | -3585.66 |
| Glyphosate | 27.51 | 0 | -27.51 |
| Gramaxone | 37.24 | -8.40 | -37.24 |
| Plastic Mulch | 1007.32 | 0 | -1007.32 |
| Waste Disposal | 287.06 | 0 | -287.06 |
| Bedding | 508.30 | 0 | -508.30 |
| Planting | 1008.26 | 0 | -1008.26 |
| Cutting, Hoeing & Hand weeding | 2706.45 | 0 | -2706.45 |
| Spraying | 117.60 | 0 | -117.60 |
| Picking (at peak time) | 24586.60 | 0 | -24586.60 |
| End of season clean up labor | 978.46 | 0 | -978.46 |
| Cooling | 3674.81 | 0 | -3674.81 |
| Packing materials | 10622.49 | 0 | -10622.49 |
| Disking | 72.04 | 0 | 0 |
| Cantaloupes seeds and transplants | 0 | 559.33 | 559.33 |
| Strawberry hand pull cost | 0 | 550.11 | 0 |
| Herbicides (Cantaloupe) | 0 | 153.34 | 153.34 |
| Fungicide & insecticide (Cantaloupe) | 0 | 2668.74 | 2668.74 |
| Fertilizer (Cantaloupe) | 0 | 593.06 | 593.06 |
| Machinery Variable cost (Cantaloupe) | 0 | 1017.61 | 1017.61 |
| Tractor: fuel cost (Cantaloupe) | 0 | 829.13 | 829.13 |
| Tractor: Lubrication cost (Cantaloupe) | 0 | 82.91 | 82.91 |
| Picking cost (Cantaloupe) | 0 | 6699.28 | 6699.28 |
| Packing cost (Cantaloupe) | 0 | 4828.95 | 4828.95 |
| Post-harvest cost (Cantaloupe) | 0 | 616.39 | 616.39 |

*(Continued)*

**Table 6.** (Continued)

| Input Names | Strawberry production costs ($/ha) | Change in costs associated with relay cropped cantaloupe ($/ha) | Differences between baseline and cantaloupe only treatments ($/ha) |
|---|---|---|---|
| Clean-up (Cantaloupe) | 0 | 168.03 | 168.03 |
| Cooling costs (Cantaloupe) | 0 | 1670.56 | 1670.56 |
| Total | 63659.72 | 20429.04 | -41197.7 |

of cantaloupes planted with and without strawberries resulted in revenue losses ranging from $250.0-$9200.0 per ha (Table 7). Mid-season planting dates resulted in higher profits solely because of yields reduction in early planting dates of cantaloupes due to plant mortality during cultural practices.

In 2017–18, all the cantaloupes planted with strawberries at five different planting dates resulted in net profits. The cantaloupes planted with strawberries on March 1 resulted in the highest profit of $14793.7/ha, following by January 4 ($10504.3/ha), January 18 ($8086.5/ha), February 1 ($7053.0/ha), and Feb 15 ($7436.1/ha). There was a significant yield difference between cantaloupes planted on January 4 and March 1. However, the net profits were not significantly different, because the market price dropped at the end of the season as cantaloupes from neighboring states come into the market. A study on relay cropping of muskmelons with scotch pine christmas tree in kansas reported to have profits of around $7800–2600 per ha based on the market price of muskmelons [39].

The Cantaloupes planted alone on January 4, and 18 resulted in profits of $5316.8 and $7737.6/ha, respectively (Table 8). All other planting dates of cantaloupes planted alone resulted in overall losses ranging from $560.0 –$8500.0 per ha. Based on two years' results, the later planting dates (February to March) resulted in higher economic returns due to higher yields. The early planting dates resulted in net losses compared to baseline treatment due to lower yields because of frost events in the early spring season.

**Table 7. The total cost and revenue comparison between strawberry productions versus strawberry relay cropped with cantaloupes in 2016–17.** A negative sign indicates reduced costs, returns and profits compared to strawberries alone.

| Treatments* | Added costs of the alternative treatment ($/ha) | Reduced returns of the alternative treatment ($/ha) | Total negative effects of the alternative treatment ($/ha) | Reduced costs of the alternative treatment ($/ha) | Added returns of the alternative treatment ($/ha) | Total positive effects of the alternative treatment ($/ha) | Net effects relative to treatment 1 ($/ha) |
|---|---|---|---|---|---|---|---|
| Only strawberries | 0.0 | 0.0 | 0.0 | 0.0 | 0.0 | 0.0 | 0.0 |
| S+C (Jan 4) | 20132.5 | 0.0 | 20132.5 | 0.0 | 16796.4 | 16796.4 | -3336.2 |
| S+ C (Jan18) | 20132.5 | 0.0 | 20132.5 | 0.0 | 16615.8 | 16615.8 | -3516.7 |
| S+ C (Feb 1) | 20132.5 | 0.0 | 20132.5 | 0.0 | 23415.3 | 23415.3 | 3282.8 |
| S+ C (Feb 15) | 20132.5 | 0.0 | 20132.5 | 0.0 | 23134.2 | 23134.2 | 3001.6 |
| S+ C (Mar 1) | 20132.5 | 0.0 | 20132.5 | 0.0 | 19867.5 | 19867.5 | -265.0 |
| C (Jan 4) | 0.0 | -66916.7 | 66916.7 | -40486.9 | 17131.3 | 57618.2 | -9298.5 |
| C (Jan18) | 0.0 | -66916.7 | 66916.7 | -40486.9 | 21412.5 | 61899.4 | -5017.3 |
| C (Feb 1) | 0.0 | -66916.7 | 66916.7 | -40486.9 | 24982.5 | 65469.4 | -1447.3 |
| C (Feb 15) | 0.0 | -66916.7 | 66916.7 | -40486.9 | 17013.4 | 57500.3 | -9416.4 |
| C (Mar 1) | 0.0 | -66916.7 | 66916.7 | -40486.9 | 17689.4 | 58176.3 | -8740.4 |

*S- Strawberry and C-Cantaloupes.
aNet effects are the profits compared to baseline (only strawberries) treatment.

**Table 8. The total cost and revenue comparison between strawberry productions versus strawberry relay cropped with cantaloupes in 2017–18.** A negative sign indicates reduced costs, returns and profits compared to strawberries alone.

| Treatments* | Added costs of the alternative treatment ($/ha) | Reduced returns of the alternative treatment ($/ha) | Total negative effects of the alternative treatment ($/ha) | Reduced costs of the alternative treatment ($/ha) | Added returns of the alternative treatment ($/ha) | Total positive effects of the alternative treatment ($/ha) | Net effects relative to treatment 1 ($/ha)[a] |
|---|---|---|---|---|---|---|---|
| Only strawberries | 0.0 | 0.0 | 0.0 | 0.0 | 0.0 | 0.0 | 0.0 |
| S+C (Jan 4) | 20132.5 | 0.0 | 20132.5 | 0.0 | 30636.8 | 30636.8 | 10504.3 |
| S+ C (Jan18) | 20132.5 | 0.0 | 20132.5 | 0.0 | 28219.0 | 28219.0 | 8086.5 |
| S+ C (Feb 1) | 20132.5 | 0.0 | 20132.5 | 0.0 | 27185.5 | 27185.5 | 7053.0 |
| S+ C (Feb 15) | 20132.5 | 0.0 | 20132.5 | 0.0 | 27568.6 | 27568.6 | 7436.1 |
| S+ C (Mar 1) | 20132.5 | 0.0 | 20132.5 | 0.0 | 34926.2 | 34926.2 | 14793.7 |
| C (Jan 4) | 0.0 | -81669.9 | 81669.9 | -40486.9 | 41963.6 | 82450.5 | 780.6 |
| C (Jan18) | 0.0 | -81669.9 | 81669.9 | -40486.9 | 44384.4 | 84871.3 | 3201.4 |
| C (Feb 1) | 0.0 | -81669.9 | 81669.9 | -40486.9 | 31388.5 | 71875.4 | -9794.5 |
| C (Feb 15) | 0.0 | -81669.9 | 81669.9 | -40486.9 | 28223.8 | 68710.7 | -12959.2 |
| C (Mar 1) | 0.0 | -81669.9 | 81669.9 | -40486.9 | 36085.8 | 76572.7 | -5097.2 |

*S- Strawberry and C-Cantaloupes.

[a]Net effects are the profits compared to baseline (only strawberries) treatment.

In conclusion, it is safe to plant a secondary crop with strawberries with no strawberry yield loss. The cost of pepper and cantaloupe production is very high, which makes these them an expensive and low-income crop to produce if planted alone depending upon planting date. However, the production of peppers and cantaloupes with strawberries as a relay crop on optimal planting dates divides the production costs between two crops. It is very crucial to optimize the date of planting of the secondary crop because relay cropping does not always result in improved economic returns [40]. However, Relay cropping of strawberries with cantaloupes and jalapeno peppers is feasible and can increase economic returns.

## Supporting information

**S1 Data.**
(XLSX)

**S2 Data.**
(XLSX)

## Acknowledgments

The author would like to thank the weed science lab members at GCREC, Balm FL.

## Author Contributions

**Conceptualization:** Nathan S. Boyd.

**Data curation:** Ravneet K. Sandhu, Qi Qiu.

**Formal analysis:** Ravneet K. Sandhu, Qi Qiu, Zhegfei Guan.

**Funding acquisition:** Nathan S. Boyd.

**Investigation:** Ravneet K. Sandhu.

**Methodology:** Ravneet K. Sandhu.

**Resources:** Nathan S. Boyd, Zhegfei Guan.

**Software:** Ravneet K. Sandhu.

**Supervision:** Nathan S. Boyd, Zhegfei Guan.

**Writing – original draft:** Ravneet K. Sandhu.

**Writing – review & editing:** Ravneet K. Sandhu, Nathan S. Boyd.

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
