## [Decision Letter · Decision Letter 0]

3 Jun 2020

PONE-D-20-11825

Optimization of planting dates of Jalapeno pepper (Capsicum annuum 'Jalapeño' L.) and cantaloupe (Cucumis melo var. cantalupensis Ser.) relay cropped with strawberry (Fragaria × ananassa Duchesne)

PLOS ONE

Dear Prof. Boyd,

Thank you for submitting your manuscript to PLOS ONE. After careful consideration, we feel that it has merit but does not fully meet PLOS ONE’s publication criteria as it currently stands. Therefore, we invite you to submit a revised version of the manuscript that addresses the points raised during the review process.

We look forward to receiving your revised manuscript.

Kind regards,

Mario Licata, Ph.D.

Academic Editor

PLOS ONE

Journal Requirements:

Additional Editor Comments (if provided):

Dear Prof. Boyd,

Your paper entitled “Optimization of planting dates of Jalapeno pepper (Capsicum annuum 'Jalapeño' L.) and cantaloupe (Cucumis melo var. cantalupensis Ser.) relay cropped with strawberry (Fragaria × ananassa Duchesne)” by Sandhu R.K. et al. is well articulated and reports scientific information on the effects of competition between pepper and cantaloupe and of planting date on crop yield. Particularly, the authors have also investigated the effects of planting on the economic return when relay cropping cantaloupe and pepper with strawberry.

I am sufficiently satisfied by the work that the authors have carried out and I believe that their paper could be published in PLOS ONE after MAJOR REVISION. In my opinion, there are some points which I would like to see improved before publication.

In the “Experimental set up” of Materials and Methods, the authors describe the type of soil of the test area. However, they do not report any information on climate characteristics. I suggest the authors to include in this part of the manuscript data about climate and, in particular, to report mean air temperature and rainfall values of the test area with reference to previous 5/10 years, please. This is fundamental to understand the environmental characteristics of the test area.

In the Materials and Methods, the authors do not report any information on irrigation and fertilization rates of the main species in the study. Please, provide.

In my opinion, discussion should be improved.

Conclusions are supported by the data but tend to repeat part of the results. In my opinion, the conclusions should be improved and authors should pay more attention on the impact of their results agriculture. I suggest the authors to improve the quality of tables and figures in agreement with the Guide for Authors of PLOS ONE. And also the reference section should be re-checked.

Reviewers' comments:

Reviewer's Responses to Questions

**Comments to the Author**

1. Is the manuscript technically sound, and do the data support the conclusions?

Reviewer #1: Yes

2. Has the statistical analysis been performed appropriately and rigorously? 

Reviewer #1: Yes

3. Have the authors made all data underlying the findings in their manuscript fully available?

Reviewer #1: Yes

4. Is the manuscript presented in an intelligible fashion and written in standard English?

Reviewer #1: Yes

5. Review Comments to the Author

Reviewer #1: The manuscript titled ‘Optimization of planting dates of Jalapeno pepper (Capsicum annuum 'Jalapeño' L.) and cantaloupe (Cucumis melo var. cantalupensis Ser.) relay cropped with strawberry (Fragaria × ananassa Duchesne)’ submitted to PLOS ONE fit with the aim of the Journal. The authors conducted four separate trials on relay cropping strawberry with jalapeño pepper and cantaloupe, in 2016-17 and 2017-18 at Balm, Florida. In the present manuscript the authors evaluated: 1) if competition between the primary and secondary crop take place in terms of yield and morphological properties, 2) the optimal planting date for the secondary crop that optimizes crop performance, 3) the effect of strawberry termination date on the yield of the secondary crop, and 4) the influence of planting date on the profits when relay cropping cantaloupes and peppers with strawberries. From my point of view, overall, the work is well written and the originality is very high since to my knowledge this is the first report on interactive effect between planting date of Capsicum annuum 'Jalapeño' L. or Cucumis melo var. cantalupensis Ser. and relay cropping with strawberry. However, the authors did not observe in a deep way the yield traits of the cultivated species such as, total production, marketable production, number of fruit and average fruit weight. Furthermore, the authors did not look on the influence of the treatments on fruit nutritional and functional properties of the vegetables and strawberry. I would like to point out that, currently, the aspects related to the effects of food on human health are highly valorised and, generally, a lot of attention is paid. I would like to stress that vegetables have recently increased in popularity by earning the title of “functional food” or “superfood”. Consequently, this might represent a weakness of the work. Nevertheless, the results of the paper are of interest for vegetable growers, agronomists and scientists.

The experimental design is solid and the statistical analysis is of quality, although improvable.

I recommend the publication of this paper after major conditions:

Title

- Line 2: “cantaluensis” should be written in italic.

Introduction

- Line 55: Instead of: ...Comopetition...it should be: .... Competition...

- Line 61: missing black

- Line 79: Instead of: …21-26 C…it should be: …21-26 °C

- Information on the effect of cultivation practices on yield and yield related traits, nutritional and functional properties of the involved vegetables and strawberry should be provided.

Please see the following referenced:

Sermenli, T., & Mavi, K. (2010). Determining the yield and several quality parameters of ‘Chili Jalapeno’in comparison to ‘Pical’and ‘Geyik Boynuzu’pepper cultivars under Mediterranean conditions. African journal of agricultural research, 5(20), 2825-2828.

Kyriacou, M. C., Leskovar, D. I., Colla, G., & Rouphael, Y. (2018). Watermelon and melon fruit quality: The genotypic and agro-environmental factors implicated. Scientia Horticulturae, 234, 393-408.

Sabatino, L., D’Anna, F., Prinzivalli, C., & Iapichino, G. (2019). Soil Solarization and Calcium Cyanamide Affect Plant Vigor, Yield, Nutritional Traits, and Nutraceutical Compounds of Strawberry Grown in a Protected Cultivation System. Agronomy, 9(9), 513.

Sabatino, L., De Pasquale, C., Aboud, F., Martinelli, F., Busconi, M., Eleonora, D. A., ... & Fabio, D. A. (2017). Properties of new strawberry lines compared with well-known cultivars in winter planting system conditions. Notulae Botanicae Horti Agrobotanici Cluj-Napoca, 45(1), 9-16.

Materials and Methods

- Line 121: Instead of …(Cucumis melo var. cantalupensis Ser.) and peppers (Capsicum annuum…it should be:…(Cucumis melo var. cantalupensis Ser.) and peppers (Capsicum annuum

- Line 123: please use the International System of Units.

- Please provide information on irrigation and fertilization management of the vegetables and strawberry. In respect to the fertilization and soil fumigation, please read and cite the recent papers such as the follow:

Sabatino, L., D’Anna, F., Prinzivalli, C., & Iapichino, G. (2019). Soil Solarization and Calcium Cyanamide Affect Plant Vigor, Yield, Nutritional Traits, and Nutraceutical Compounds of Strawberry Grown in a Protected Cultivation System. Agronomy, 9(9), 513.

- Since the experiments were repeated twice, in the ANOVA analysis, please include the year as fix factor.

- Please add, also, a heat map (please see the following link https://biit.cs.ut.ee/clustvis/), which might provide an extra value to the work.

Results and Discussion

- Please re-write the paragraph accordingly to the revised statistical analysis (ANOVA analysis and heat map).

- Please, try to explain in a deeper way the results. I miss some speculations concerning some physiological processes involved…

Conclusions

- Line 366: missing black

- Line 370: Instead of … alone Unlike…it should be… alone. Unlike

- Line 371: Instead of … without yiesld loss…it should be… without yield loss…

Based on the above considerations I recommend the Editor to accept the manuscript for publication in PLOS ONE after the aforementioned revisions.

6. PLOS authors have the option to publish the peer review history of their article (what does this mean?). If published, this will include your full peer review and any attached files.

Reviewer #1: No

---

## [Author Response · Author response to Decision Letter 0]

29 Jun 2020

Please see the response to review document

---

## [Decision Letter · Decision Letter 1]

13 Jul 2020

Optimization of planting dates of Jalapeno pepper (Capsicum annuum 'Jalapeño' L.) and cantaloupe (Cucumis melo var. cantalupensis Ser.) relay cropped with strawberry (Fragaria × ananassa Duchesne)

PONE-D-20-11825R1

Dear Dr. Boyd,

We’re pleased to inform you that your manuscript has been judged scientifically suitable for publication and will be formally accepted for publication once it meets all outstanding technical requirements.

Kind regards,

Mario Licata, Ph.D.

Academic Editor

PLOS ONE

Additional Editor Comments (optional):

Dear Dr Boyd,

I am pleased to inform you that the paper entitled “Optimization of planting dates of Jalapeno pepper (Capsicum annuum 'Jalapeño' L.) and cantaloupe (Cucumis melo var. cantalupensis Ser.) relay cropped with strawberry (Fragaria × ananassa Duchesne)" (Ref. No.: PONE-D-20-11825R1) can be accepted in its present form for publication in PLOS ONE. All the requested corrections were made in the revised manuscript by the authors and I am satisfied by their work.

Kind regards,

The Academic Editor

Reviewers' comments:

Reviewer's Responses to Questions

**Comments to the Author**

1. If the authors have adequately addressed your comments raised in a previous round of review and you feel that this manuscript is now acceptable for publication, you may indicate that here to bypass the “Comments to the Author” section, enter your conflict of interest statement in the “Confidential to Editor” section, and submit your "Accept" recommendation.

Reviewer #1: All comments have been addressed

2. Is the manuscript technically sound, and do the data support the conclusions?

Reviewer #1: Yes

3. Has the statistical analysis been performed appropriately and rigorously? 

Reviewer #1: Yes

4. Have the authors made all data underlying the findings in their manuscript fully available?

Reviewer #1: Yes

5. Is the manuscript presented in an intelligible fashion and written in standard English?

Reviewer #1: Yes

6. Review Comments to the Author

Reviewer #1: (No Response)

7. PLOS authors have the option to publish the peer review history of their article (what does this mean?). If published, this will include your full peer review and any attached files.

Reviewer #1: No

---

## [Editor Report · Acceptance letter]

15 Jul 2020

PONE-D-20-11825R1 

Optimization of planting dates of Jalapeno pepper (Capsicum annuum 'Jalapeño' L.) and cantaloupe (Cucumis melo var. cantalupensis Ser.) relay cropped with strawberry (Fragaria × ananassa Duchesne) 

Dear Dr. Boyd:

I'm pleased to inform you that your manuscript has been deemed suitable for publication in PLOS ONE. Congratulations! Your manuscript is now with our production department. 

Kind regards, 

on behalf of

Dr. Mario Licata 

Academic Editor

PLOS ONE